# Antibiotic Stewardship in *Staphylococcus aureus* Bloodstream Infection Treatment—Analysis Based on 29,747 Patients from One Hospital

**DOI:** 10.3390/antibiotics9060338

**Published:** 2020-06-18

**Authors:** Grzegorz Ziółkowski, Iwona Pawłowska, Estera Jachowicz, Michał Stasiowski

**Affiliations:** 1Sosnowiec Medical College, 41-200 Sosnowiec, Poland; nc3@wp.pl; 2Division of Microbiology and Epidemiology, St. Barbara Specialized Regional Hospital No. 5, 41-200 Sosnowiec, Poland; ivi5@op.pl; 3Department of Microbiology, Faculty of Medicine Jagiellonian University Medical College, 31-007 Krakow, Poland; 4Clinical Department of Anaesthesiology and Intensive Therapy, Faculty of Medical Sciences in Zabrze, Medical University of Silesia, 40-055 Katowice, Poland; mstasiowski.anest@gmail.com; 5Department of Anaesthesiology and Intensive Therapy, St. Barbara’s Memorial Regional Hospital in Sosnowiec, pl. Medyków 1, 41-200 Sosnowiec, Poland

**Keywords:** *Staphylococcus aureus*, bloodstream infections, MRSA, MSSA

## Abstract

Some of the most serious healthcare-associated infections (HAI) are highly deadly bloodstream infections (BSIs) caused by *Staphylococcus aureus*. The aim of the study was to analyse compliance of treatment practice with clinical guidelines in patients with *S. aureus* BSIs. The study was conducted at the Sosnowiec Hospital, Poland in 2019. During the study, 29,747 patients were hospitalized and 41 *S. aureus* BSIs (only HAIs) episodes were observed. According to local clinical practice guidelines, each case of BSI required blood cultures, echocardiography and control culture after the implementation of the targeted therapy. Incidence rate of *S. aureus* BSI was 0.8/1000 admissions; the greatest department admission rates were in the ICU (19.3/1000 admissions) and in the Nephrology Department (8.7/1000 admissions). Only 2 patients were treated following the protocol (4.8%); the most common errors were the use of an inappropriate drug or incorrect duration of antibiotic treatment. No patient underwent echocardiography, and control cultures were performed in 70% of cases. The case fatality rate was 7.3%. A satisfactorily low case fatality rate was found despite the poor antibiotic stewardship. Lack of discipline concerning antibiotic use can strongly impact the observed high drug resistance in HAIs and high *Clostridioides difficile* incidence rate in the studied hospital.

## 1. Introduction

Modern medicine gives us a possibility to treat diseases that, until now, have been incurable or untreated, but it also poses a huge risk of developing infections, which subsequently entail intensive antimicrobial treatment that can cause an adverse reaction, including bacterial acquisition of resistance to antibiotics. Drug-resistant microorganisms that spread easily include the methicillin-resistant *Staphylococcus aureus* (MRSA). MRSA is a serious problem in southern Poland, as its prevalence in healthcare-associated infections (HAI) is 15.1% in total, and in invasive infections (only HAIs), it is as much as 26.7% [1]. 

Bloodstream infections (BSIs) are some of the most dangerous HAIs, and those caused by *Staphylococcus aureus* are especially dangerous. They amount to 13.4% of all BSIs [2] in intensive care units (ICUs) of southern Poland and, in our hospital, they are associated with 4.6% [3]. *S. aureus* BSI involves a high risk of death: in the USA in 2005–2016, the in-hospital mortality among patients with *S. aureus* BSI was 18%, but mortality rates were higher for HAIs: 29% for MRSA and 24% for MSSA (methicillin-sensitive *Staphylococcus aureus*) [4]. In Poland, in-hospital mortality among patients with HA-BSI was 20% [2]. The EARS-Net (European Antimicrobial Resistance Surveillance Network) database, which is based on data from routine clinical studies on antimicrobial resistance from local laboratories, works also in Poland. In 2018, the EU/EEA population-weighted mean percentage of methicillin-resistant *S. aureus* (MRSA) was 16.4%. The highest prevalence of MRSA was isolated in southern Europe, e.g., in Italy (34%), Portugal (38.1%) and Romania (43%). The prevalence of MRSA in Poland in 2018 was 15.8%. The results of a European Centre for Disease Prevention and Control report significantly showed that, in Europe, the level of MRSA in invasive infections had a decreasing trend between 2015 and 2018 [5]. In addition, the European Surveillance of Antimicrobial Consumption Network (ESAC-Net) operates in Poland. ESAC-Net is a pan-European network that contains data on the use of antimicrobials in the community and in the hospital sector.

For several years now, there have been intensive efforts to develop standards for the use of antibiotics to reduce the risk of spreading drug resistance. The Antibiotic Stewardship Program (ASP) involves coordinated actions consisting of promoting the principles of rational antibiotic use—from the selection of an optimal antibiotic, through the choice of an appropriate dose and duration of treatment, to the route of administration [6]. A vital aspect and one of the key elements of an ASP in clinical practice is the use of data, including the results of microbiological testing and others. Unfortunately, the importance of microbiological diagnostics seems to be underestimated by Polish doctors [7,8]. Conducting continuous education and postgraduate studies for medical staff seems to be very necessary.

With regard to *S. aureus* BSIs, the elements concerning diagnosis and treatment of infections that are of essential importance with respect to the effectiveness of therapy were described in detail. These include consultation with a specialist in infectious diseases, appropriate administration of the antibiotic along with proper therapy duration, and monitoring the effectiveness of antibiotic therapy using control blood cultures and echocardiography (transesophageal or transthoracic) in order to diagnose endocarditis. When these actions are undertaken concurrently, they significantly reduce the mortality associated with BSI-SA [9,10,11,12,13]. Therefore, all of these activities are also components of a hospital ASP, which is closely related not only to the improvement in patient safety and supervision of antibiotic consumption, but also to the reduction of drug resistance [6].

Analysis of the effectiveness of therapy for *S. aureus* BSIs is the basis of infection surveillance [14] and, since modern medicine also involves modern microbiology and other diagnostic tests, which provide key data necessary for effective treatment, the actions conducted in the field of surveillance and ASP should encompass, apart from the use of antibiotics (appropriate drug, appropriate dose and appropriate time of administration), the inspection of treatment correctness as well. The authors’ experiences associated with high drug resistance in infections and, not less importantly, high antibiotic consumption had already been published [3], however, they concerned the general situation and one department only, the ICU. Hence, our hospital saw a need for a critical look at the quality of treatment of BSIs caused by *Staphylococcus aureus*. The aim of this study was a retrospective analysis of the course of targeted antibiotic therapy for laboratory-confirmed bloodstream infections with *Staphylococcus aureus* taking into consideration the varying sensitivity of the strains and the characteristics of the patients treated. 

## 2. Materials and Methods

It was a retrospective laboratory-based study of healthcare-associated *S. aureus* BSIs in patients treated in 2019 at the St. Barbara Specialised Regional Hospital No. 5, Sosnowiec., Poland. The institution is a non-teaching secondary care hospital and one of the biggest institutions in the south of Poland with 652 beds treating medical, surgical and trauma patients in 10 surgical units and 8 medical units, including a cardiology department. At the hospital, there is a 16-bed intensive care unit. Patients are admitted to the hospital both electively and in an emergency through the Emergency Unit. Emergency admissions constituted around 24.5% of all admissions. In the period under study, 29,747 patients were hospitalised, and the number of person-days spent in the hospital totalled 133,002.

In the event of suspected BSI, two or more sets of blood cultures were drawn under aerobic and anaerobic conditions; the material was taken simultaneously from different sites and repeated if necessary. The blood samples were cultured using the BACTEC FX 40 Automated Blood Culture System (Becton Dickinson, Warszawa, Poland) for 5 days. Only non-repetitive *S. aureus* isolates were selected for microbiological analysis. Further cultures from the same patient and the same case of LC-BSI (laboratory-confirmed bloodstream infection) were excluded from examination. Antibiotic susceptibility testing was carried out using the Phoenix 100 automated system (Becton Dickinson, Warszawa, Poland). Drug susceptibility was identified and determined with the use of Combo panels: PMIC/ID and criteria for interpretation in accordance with EUCAST v.9.0 [15]. Sensitivity to methicillin was determined by the disk diffusion method using a 30 µg cefoxitin disk [10]. On the basis of the assessment of drug susceptibility of the strain under study to cefoxitin, the methicillin resistance phenotype was established, i.e., either MSSA or MRSA. The result of microbiological testing (blood culture) with antibiogram was available to the attending physician after 72 h, at the latest. 

HA *S. aureus* BSIs were classified as primary (central venous catheter-related, CVC-BSI, or not) and secondary, respectively, into pneumonia S-PUL, urinary tract infection S-UTI and surgical site infection S-SSI, according to the definitions of the ECDC (European Centre for Disease Prevention and Control) [8]. Clinical, radiological and microbiological evidence was used to determine the source of secondary BSIs. In one CVC-BSI case, the blood culture was analysed in parallel with the culture from the CVC tip, and both tests gave a positive result showing growth of the same microorganism. In the remaining CVC-BSI cases, the removed vascular catheters were not examined microbiologically. None of the patients with symptoms of S-SSI were reoperated upon. In three patients, *S. aureus* was obtained from biopsy cultures (orthopaedic and neurosurgery), and in other cases, it was done from wound swabs. 

The basis for the analysis was a unified Treatment Quality Assessment Card, filled in by the attending physician, which contained data on:the appropriate antibiotic, its dose and duration of treatment,conducting blood cultures: the first one and controls after 72 h from the beginning of treatment, andechocardiography, the decision on whether to execute it was made by a cardiologist in agreement with the attending physician.

The hospital formulary pointed to Cloxacillin (Syntarpen, Polfa Tarchomin, Poland) as the first-line drug for MSSA-BSI.

The exclusion criteria included community-acquired BSIs, polymicrobial cultures or positive cultures that were deemed to be contaminated. The hospital does not employ a medical microbiologist or an infectious disease physician. 

For statistical analysis of ordinal numbers or dichotomous data, information on the number and percentage of people was used. The average and standard deviation (SD) from 95% confidence interval (CI), minimum and maximum were calculated. Statistical significance was assumed at *p* < 0.05. Statistical analysis of the data was carried out using the SPSS software (SPSS—Statistical Package for the Social Sciences, STATISTICS 24, Armonk, NY, USA). 

## 3. Results

41 BSI episodes of *S. aureus* (28 men and 13 women) were included in the study; the average age of the surveyed people was 64 years, SD 14, 95% CI, 1.46, minimum of 34 years, and maximum of 97 years. The incidence rate was 0.8 per 1000 admissions and 3.1/10,000 person-days of hospital stay. The majority of *S. aureus* BSI cases were reported in the Nephrology Department (8, 19.5% of all) and in the Department of Internal Medicine (6, 14.6% of all). The greatest incidences were associated with the Intensive Care Unit—19.3/1000 admissions and Nephrology—8.7/1000 admissions (Table 1).

Most of the *S. aureus* BSI cases were primary, including 3 CVC-BSI cases (58.5% of all). Secondary infections were dominated by S-PULs and S-UTIs, both 7 cases (17.1% of all) (Table 2). 

MRSA drug resistance concerned 2 strains, 4.8% of all studied patients; they were CVC-BSI or S-PUL (Table 1).

Pursuant to the protocol, 2 patients (5.12%) were treated: CVC-BSI (in the orthopaedic ward) and S-PUL (ICU). Targeted antimicrobial chemotherapy for MSSA-BSI patients:in 1 patient (2.5% of all), cloxacillin was administered at the recommended dose and the treatment duration followed the procedure: 14 days.in 5 patients (12.8% of all), cloxacillin was administered at the recommended dose, but the duration of treatment was too short and inconsistent with the recommended treatment duration, i.e. it was from 3 to 10 days.in 33 patients (84,61% of all), targeted therapy was conducted in breach of the procedure:○in 17 cases, cefuroxime or amoxicillin/clavulanic acid for 3–6 days were used○8 patients received carbapenem for 3 days○4 patients were given ampicillin + sulbactam for 10 days○and the remaining patients received ciprofloxacin plus metronidazole for 12 days.

MRSA-BSI targeted treatment:vancomycin according to the recommended dose and duration of treatment—1 patient (2.4% of all),and in the second case, imipenem was applied.

3 people died in the course of *S. aureus* BSI treatment. The case fatality rate was 7.3%. The immediate causes of death, according to the doctor’s qualification, were respiratory and circulatory failures, i.e.:a patient with alcoholic cirrhosis and ascites, treated (only empirically) with cipronex plus metronidazole, died on day 2 of empirical treatment;a patient with staphylococcal sepsis, treated (only empirically) with amoksiklav, died after 6 h of empirical treatment;and a patient with left-sided pneumonia and staphylococcal sepsis, treated (only empirically) with cefuroxime, died on day 3 of empirical treatment.

Deaths were associated only with MSSA infections. 

None of the patients underwent echocardiography. Control cultures were carried out in 30 patients (73.1%) from the following wards: ICU, neurosurgery, orthopaedics, pulmonology, internal medicine, gastroenterology, nephrology, neurology, diabetology, urology, and dialysis.

## 4. Discussion

The data in this study indicate significant problems with regard to practising ASP tasks. Only in relation to 5.12% of cases were appropriate therapeutic regimens employed, i.e., the right drug and duration of treatment, and only in 73% of cases was the effectiveness of treatment checked using microbiological testing of blood cultures, and finally, none of the patients underwent additional examination—that is, echocardiography. Two cases of SA-BSI were diagnosed in the orthopaedics ward, and one (50%) of them was treated correctly. In contrast, 5 cases of SA-BSI were diagnosed in the ICU—only 1 case (20%) was treated correctly.

Hence, the overall compliance with ASP procedures was 0%.

The source of *S. aureus* BSIs can be both severe and life-threatening inflammatory foci such as infective endocarditis, epidural abscesses in the lumbar spine, abscesses of the liver and spleen, infections of bones, including spinal vertebrae [1,3,4,6,7], as well as the application of vascular lines, including central lines. Despite the availability of guidelines concerning the treatment of a *S. aureus* BSI, its 30-day mortality remains at a high level of 15–40%, depending on the patient’s age, clinical symptoms and comorbidities [2,8,14]. One of the reasons behind this state of affairs may be the fact that a large proportion of LC-BSI cases are secondary BSIs. In the present data, the proportion of primary BSIs lived up to expectations, as it was 58.5%, while Kaech et al. report 52%, [16], and Cuijpers et al. 40% [17]. However, the dominance of secondary BSIs may result in difficulties with regard to choosing the appropriate empirical therapy and then targeted therapy, as, from the point of view of microbiological diagnostics, the test result does not serve to unequivocally diagnose the infection, though it supports making the right clinical decisions. Therefore, it is so vital in clinical practice to make use of additional information, e.g. therapeutic success in the treatment of *S. aureus* BSI is greatly improved through consultation with an infectious disease specialist, appropriate antibiotic therapy, monitoring the effectiveness of treatment using control blood cultures and the employment of (transesophageal or transthoracic) echocardiography in order to diagnose endocarditis [9,10,11,12,13]. Transthoracic echocardiography is particularly indicated in patients with an embolic episode, history of bacterial endocarditis, abscess of the spine or spinal canal, and patients with osteomyelitis [12,13].

Therapeutic success in the treatment of *S. aureus* BSI depends, among others, on the duration of antibiotic therapy and the antibiotic chosen [18]. In our study, the most frequent errors in the course of pharmacotherapy were its too short duration and/or the application of an inappropriate antibiotic, i.e., one that is not recommended for the treatment of HAIs. The most disturbing fact in the present data is shortening the duration of antibiotic therapy. The problem concerned 95% of cases. Such actions can lead to increased microbial resistance to antibiotics. In their previous studies, the authors pointed out the problem of multidrug resistance (MDR) in the hospital in Sosnowiec: the prevalence of MDR in SSI infections amounted to 22.6%, while for Gram-negative rods, it was significantly higher: 97.6% for *Acinetobacter baumannii* and 50.0% for *Klebsiella pneumoniae* [19]. In another study, 95% of strains isolated from the blood drawn from patients with pneumonia were resistant to imipenem and meropenem, and 100% to cephalosporins and tetracyclines [3]. Moreover, yet another study [20] demonstrated that almost half (42.8%) of the *P. aeruginosa* strains tested were identified as MBL-producing (metallo-beta-lactamase producing) strains, 85.7% were meropenem-resistant, 14.2% were MDR and as many as 38% were classified as XDR (extensively drug resistant). These studies indicate serious clinical problems and emphasize the need for rational treatment.

Equally significant is the matter of both inappropriate and unnecessary application of broad-spectrum antibiotics. Rhee et al. point to a potential link between this type of therapy and increased mortality [21]. The majority of patients with BSI are not infected with resistant pathogens but are still often given broad-spectrum antibiotics [21]. This is unfortunately confirmed by the present data—only 4.8% of the examined strains were highly drug resistant and did not require the administration of drugs other than the ones in the pharmacopoeia. The application of broad-spectrum antibiotics is associated with the risk of adverse effects such as *C. difficile* infections or kidney failure [21]. It is corroborated by prior reports by the authors as, in the hospital under study, the incidence rate of *C. difficile* was significantly high and even reached 2.4% in the ICU [3].

Additional tests were performed very rarely, while it is reported by other authors that it is possible to obtain much higher compliance, e.g., in the studies by Saunderson et al., control blood cultures were carried out in 93% of patients and echocardiography in 91% of patients [14]. Transthoracic echocardiography, performed in order to exclude latent endocarditis [22], should be carried out in all patients, even those at low risk of endocarditis [23]. It is sometimes considered too aggressive for elderly people as it is an invasive procedure. Accordingly, in a study by Durante-Mangoni et al., fewer surgical treatments were demonstrated in elderly patients with infective endocarditis and higher mortality was found compared to younger patients [24]. And optimization of clinical care practices is especially important, particularly for the oldest patients, to improve their prognosis in the treatment of *S. aureus* BSI [25].

A retrospective study by Kawasuji et al. analysed the relationship between the application of appropriate guidelines for the diagnosis and therapy of MRSA BSI and the availability of consultations with an infectious disease specialist. It was found that patients who received such consultations were more often treated according to expectations, i.e. the therapy was long enough (over 14 days) and employed echocardiography and control blood cultures. Furthermore, the patients who received early ID consultation more often received appropriate empirical therapy, had a total mortality rate that was three times shorter and a required shorter hospital stay, among other things [26].

The observed cases of LC-BSI did not deviate from the profile of the patient with LC-BSI. They were mainly men aged over 65, which is confirmed by the literature [7,27,28,29]. On the other hand, the mortality amounting to 7.3% was not high and only concerned very fast deaths, in which empirical treatment lasted several, up to over a dozen, hours. The results of the analysis indicate the need to implement more sensitive and faster methods of microbiological diagnostics (NAAT) as a first step of reducing waiting time due to faster identification of the etiological factor for BSI. NAAT shortens the time of identification of microorganisms without culture [30,31]. These diagnostic techniques are still very rarely used in Polish hospitals. They are considered as expensive procedures by hospital management. This was the first study that assessed the compliance of the implemented ASP procedure; the results indicate that ASPs are very needed. The effectiveness of educational activities in the fight against antimicrobial resistance has already been demonstrated. In particular, it seems to be crucial to develop appropriate curricula for teaching medical and nonmedical undergraduate students about among others judicious antibiotic prescribing [32,33].

## 5. Limitations

The physicians were trained in completing questionnaire on the quality of *S. aureus* bloodstream infections treatment, but the procedure was not validated.

## 6. Conclusions

To enhance the quality of *S. aureus* BSI therapy, it is recommended to carry out control blood cultures after 72 h of treatment, conduct ongoing microbiological consultation, perform echocardiography and implement effective treatment regimens that are recommended—using the appropriate drug at the right dose and for the proper amount of time. Unfortunately, the implementation of these rules into everyday practice can prove difficult and, in the hospital under study, it failed. Lack of data from both Poland and other countries of the region makes it impossible to conduct comparative analysis, hence, further research is needed into the degree of implementation of effective ASP programmes in Polish hospitals. The results indicate the need for interventions that should be implemented, at the organizational or individual level, to improve the patient safety and practice in infections treatment.

## Figures and Tables

**Table 1 antibiotics-09-00338-t001:** *Staphylococcus aureus* bloodstream infections in Sosnowiec Hospital, Poland in 2019.

Ward	Number of Admissions	*S.aureus* BSI * Incidence Rate	Number of Cases		
			MSSA	MRSA	Total
Intensive Care Unit	259	19.3	4	1	5
Department of Clinical Neurosurgery	1 312	3.8	5	0	5
Orthopaedic Surgery	1747	0.6	1	0	1
Department of Internal Medicine	1 468	4.1	6	0	6
Department of Gastroenterology	1 563	1.3	2	0	2
Department of General Cardiology	3 118	0.3	1	0	1
Department of Nephrology	924	8.7	8	0	8
Department of Neurology	690	5.8	3	1	4
Department of Urology	1 903	0.5	1	0	1
Department of Pulmonology	999	2.0	2	0	2
Department of Cardiac Rehabilitation	278	3.6	1	0	1
Department of Diabetology	673	1.5	1	0	1
Extracorporeal Dialysis Unit	11 013	0.3	3	0	3
Accident and Emergency Department	24 122	0.04	1	0	1
TOTAL	50 069	0.82	39	2	41

* per 1000 admissions.

**Table 2 antibiotics-09-00338-t002:** Sources of bloodstream infections.

Bloodstream Infections, Primary	Number
Primary infection	21 (51.2%)
Associated with the central vascular catheter	3 (7.3%)
Bloodstream infections secondary to:	
Pneumonia, S-PUL	7 (17.1%)
Urinary Tract Infection, S-UTI	7 (17.1%)
Surgical Site Infection, S-SSI	3 (7.3%)
Total	41 (100%)

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
