# Peer review of "Antibiotic Stewardship in Staphylococcus aureus Bloodstream Infection Treatment—Analysis Based on 29,747 Patients from One Hospital"

_antibiotics, 2020, doi:10.3390/antibiotics9060338_

Round 1

Reviewer 1 Report

The work seems interesting but there are several parts which need improvements. In particular, the methods and the results.

Major revisions:

In the methods there is no mention to any of the aspects related to the statistics which was used.

In the results section, all the times Authors mention, for example, means they should also introduce Standard deviation. In addition, they should also add CI when they have estimates.

In the Discussion section, Authors should include the comparisons with inferential aspects.

Minor revisions:

Some points to improve:

  • Line 24: Point after observed.
  • Enter European epidemiological data and programs for antibiotics and antibiotic resistance management if available.
  • Secondary infections to pneumonia are indicated as S-PUL and then as P-PNU.
  • Review the list of MSSA-BSI patient treatments. There should be 39 patients but the sum is 40. So check the results and discussions based on these numbers.
  • I suggest that in the introduction and in the conclusions we go deeper into the question of the importance of multidisciplinary educational programs for health care professionals and surveillance programs.

In this regard, I suggest reading the following article:

Fattorini M, Rosadini D, Messina G, Basagni C, Tinturini A, De Marco MF. A multi-disciplinary educational programme for the management of a carbapenem-resistant Klebsiella pneumoniae outbreak: an Italian experience. J Hosp Infect. 2018;99(4):427-428. doi:10.1016/j.jhin.2018.04.006

Author Response

DETAILED RESPONSE TO REVIEWERS:

STEP-BY-STEP REPLIES TOREVIEWERS' COMMENTS:

Reviewer #1:

The work seems interesting but there are several parts which need improvements. In particular, the methods and the results.

Authors’ reply: Thank you for this comment!

Major revisions: In the methods there is no mention to any of the aspects related to the statistics which was used.

Authors’ reply: Corrected according to suggestions (lines 129-133).

In the results section, all the times Authors mention, for example, means they should also introduce Standard deviation. In addition, they should also add CI when they have estimates.

Authors’ reply: Corrected according to suggestions (lines 135-137).

In the Discussion section, Authors should include the comparisons with inferential aspects.

Authors’ reply: Corrected according to suggestions (lines 248-249).

Minor revisions: Some points to improve: Line 24: Point after observed.

Authors’ reply: Corrected according to suggestions.

Enter European epidemiological data and programs for antibiotics and antibiotic resistance management if available.

Authors’ reply: Corrected according to suggestions (lines 49-54).

Secondary infections to pneumonia are indicated as S-PUL and then as P-PNU.

Authors’ reply: Corrected according to suggestions.

Review the list of MSSA-BSI patient treatments. There should be 39 patients but the sum is 40. So check the results and discussions based on these numbers.

Authors’ reply: Corrected according to suggestions.

I suggest that in the introduction and in the conclusions we go deeper into the question of the importance of multidisciplinary educational programs for health care professionals and surveillance programs. In this regard, I suggest reading the following article: Fattorini M, Rosadini D, Messina G, Basagni C, Tinturini A, De Marco MF. A multi-disciplinary educational programme for the management of a carbapenem-resistant Klebsiella pneumoniae outbreak: an Italian experience. J Hosp Infect. 2018;99(4):427-428. doi:10.1016/j.jhin.2018.04.006

Authors’ reply: Corrected according to suggestions (lines 62-63, 248-252, 265-266).

Reviewer 2 Report

  1. This is a short article stating q satisfactorily low case fatality rate despite poor the antibiotic treatment.
  2. I don't think it qualifies for a full length research article, may be a communications would be appropriate type.
  3. Some references seems old like 15 years back" Fowler, V.; Olsen, M.; Corey, R. Woods, C.W.; Cabell, C.H.; Clinical identifiers of complicated 283 Staphylococcus aureus bacteremia. Arch Intern Med 2003, 163, 2066-72", can authors update them.
  4. There can be discrepancy in Treatment Quality Assessment Card filling, which can be a limitation for analysis, would be better if authors can incorporate a small paragraph stating limitations of the study.

The rest of findings seems appropriate for endorsement and as mentioned above study does not suit a full length original article, instead a short communication or a study report would be appropriate. 

Author Response

DETAILED RESPONSE TO REVIEWERS:

STEP-BY-STEP REPLIES TOREVIEWERS' COMMENTS:

Reviewer #2:

This is a short article stating q satisfactorily low case fatality rate despite poor the antibiotic treatment. I don't think it qualifies for a full length research article, may be a communications would be appropriate type.

Authors’ reply: Thank you for this comment! According to MDPI website “Original research manuscripts. The journal considers all original research manuscripts provided that the work reports scientifically sound experiments and provides a substantial amount of new information. Authors should not unnecessarily divide their work into several related manuscripts, although Short Communications of preliminary, but significant, results will be considered. Quality and impact of the study will be considered during peer review.”

Some references seems old like 15 years back" Fowler, V.; Olsen, M.; Corey, R. Woods, C.W.; Cabell, C.H.; Clinical identifiers of complicated 283 Staphylococcus aureus bacteremia. Arch Intern Med 2003, 163, 2066-72", can authors update them.

Authors’ reply: Corrected according to suggestions, the reference is corrected, we used: https://www.eucast.org/fileadmin/src/media/PDFs/EUCAST_files/Disk_test_documents/2020_manuals/Manual_v_8.0_EUCAST_Disk_Test_2020.pdf.

There can be discrepancy in Treatment Quality Assessment Card filling, which can be a limitation for analysis, would be better if authors can incorporate a small paragraph stating limitations of the study.

Authors’ reply: Corrected according to suggestions (lines 253-255).

The rest of findings seems appropriate for endorsement and as mentioned above study does not suit a full length original article, instead a short communication or a study report would be appropriate. 

Authors’ reply: Thank you for this comment! According to MDPI website “Original research manuscripts. The journal considers all original research manuscripts provided that the work reports scientifically sound experiments and provides a substantial amount of new information. Authors should not unnecessarily divide their work into several related manuscripts, although Short Communications of preliminary, but significant, results will be considered. Quality and impact of the study will be considered during peer review.”

Round 2

Reviewer 1 Report

The work seems interesting but there are several parts which need improvements. In particular, the methods and the results.

Authors’ reply: Thank you for this comment!

  • Thanks for your answers.

Major revisions: In the methods there is no mention to any of the aspects related to the statistics which was used.

Authors’ reply: Corrected according to suggestions (lines 129-133).

  • In the methods, there is "average median (Me)". Clarify what has been done, please. In the results, the average is reported. 

In the results section, all the times Authors mention, for example, means they should also introduce Standard deviation. In addition, they should also add CI when they have estimates.

Authors’ reply: Corrected according to suggestions (lines 135-137).

  • Explain better in methods, please.

In the Discussion section, Authors should include the comparisons with inferential aspects.

Authors’ reply: Corrected according to suggestions (lines 248-249).

  • Try to extend the statistical analysis in results and discussions if possible. 

Minor revisions: Some points to improve: Line 24: Point after observed.

Authors’ reply: Corrected according to suggestions.

  • OK

Enter European epidemiological data and programs for antibiotics and antibiotic resistance management if available.

Authors’ reply: Corrected according to suggestions (lines 49-54).

  • Maybe more data could have been reported and discussed.

Secondary infections to pneumonia are indicated as S-PUL and then as P-PNU.

Authors’ reply: Corrected according to suggestions.

  • Line 109 is specified and used S-PUL according to ECDC definitions and also in Table 2. Line 144 and 146 are used S-PNU and PNU-BSI without having previously specified their meaning. Specify these if different from s-pul or always use the same abbreviation for easier reading. 

Review the list of MSSA-BSI patient treatments. There should be 39 patients but the sum is 40. So check the results and discussions based on these numbers.

Authors’ reply: Corrected according to suggestions.

  • Line 157 was one point among 33 patients. It' out now. Fix it, please 

I suggest that in the introduction and in the conclusions we go deeper into the question of the importance of multidisciplinary educational programs for health care professionals and surveillance programs. In this regard, I suggest reading the following article: Fattorini M, Rosadini D, Messina G, Basagni C, Tinturini A, De Marco MF. A multi-disciplinary educational programme for the management of a carbapenem-resistant Klebsiella pneumoniae outbreak: an Italian experience. J Hosp Infect. 2018;99(4):427-428. doi:10.1016/j.jhin.2018.04.006

Authors’ reply: Corrected according to suggestions (lines 62-63, 248-252, 265-266).

  • Okay, but the reference 31 contains errors in the names. Please correct reference 31.

Author Response

Thank for yours  comments.

  • In the methods, there is "average median (Me)". Clarify what has been done, please. In the results, the average is reported. 
  • Authors’ reply Corrected

  • Explain better in methods, please.
  • Authors’ reply Corrected according to suggestions (lines 132-140).
  • Try to extend the statistical analysis in results and discussions if possible. 
  • Authors’ reply Corrected according to suggestions (lines 182-184)

  • Maybe more data could have been reported and discussed.
  • Authors’ reply Corrected according to suggestions ( lines 52-57)

  • Line 109 is specified and used S-PUL according to ECDC definitions and also in Table 2. Line 144 and 146 are used S-PNU and PNU-BSI without having previously specified their meaning. Specify these if different from s-pul or always use the same abbreviation for easier reading. 
  • Authors’ reply: Corrected according to suggestions.
  • Line 157 was one point among 33 patients. It' out now. Fix it, please 
  • Authors’ reply: Corrected according to suggestions. Percentages were also recalculated. We apologize for our error in calculations
  • the reference 31 contains errors in the names. Please correct reference 31.

Authors’ reply: Corrected